# Phonon-assisted up-conversion photoluminescence of quantum dots

Zikang Ye [1,4], Xing Lin [1,2,4], Na Wang[1], Jianhai Zhou[3], Meiyi Zhu[1], Haiyan Qin [1✉] & Xiaogang Peng [1✉]

Phonon-assisted up-conversion photoluminescence can boost energy of an emission photon to be higher than that of the excitation photon by absorbing vibration energy (or phonons) of the emitter. Here, up-conversion photoluminescence power-conversion efficiency (power ratio between the emission and excitation photons) for CdSe/CdS core/shell quantum dots is observed to be beyond unity. Instead of commonly known defect-assisted up-conversion photoluminescence for colloidal quantum dots, temperature-dependent measurements and single-dot spectroscopy reveal the up-conversion photoluminescence and conventional down-conversion photoluminescence share the same electron-phonon coupled electronic states. Ultrafast spectroscopy results imply the thermalized excitons for up-conversion photoluminescence form within 200 fs, which is 100,000 times faster than the radiative recombination rate of the exciton. Results suggest that colloidal quantum dots can be exploited as efficient, stable, and cost-effective emitters for up-conversion photoluminescence in various applications.

[1] Key Laboratory of Excited-State Materials of Zhejiang Province, Department of Chemistry, Zhejiang University, Hangzhou 310027, China. [2] College of Information Science and Electronic Engineering, Zhejiang University, Hangzhou 310027, China. [3] Najing Technology Corporation LTD, Hangzhou 310056, China. [4] These authors contributed equally: Zikang Ye, Xing Lin. ✉email: hattieqin@zju.edu.cn; xpeng@zju.edu.cn

Photo- and electro-excited light emission (photo-luminescence and electroluminescence) play an essential role in our daily life, especially in information presentation, acquisition, and communication. These applications currently employ down-conversion light emission, with the energy of the emitting photons significantly lower than that of the excitation photons (or electron–hole pairs). For general lighting alone, down-conversion light emission consumes nearly 15% of global electric power and causes serious energy loss in the form of waste heat[1]. Waste heat shortens device lifetime, raises fabrication cost, contributes significantly to the global warming, and causes safety concerns. Though less known, thermal assisted up-conversion photoluminescence (UCPL) can emit a photon with energy higher than the excitation photon by extracting thermal energy stored as molecular vibration in the working media (or phonons in crystals).

Rare-earth-doped materials show efficient UCPL[2–4], but they are sophisticated in preparation, low in absorption cross-section, and difficult to be integrated into optoelectronic devices. Though optical cooling—samples being cooled upon sub-bandgap photon excitation—based on UCPL of organic dyes dissolved in solution was reported about 25 years ago[5,6], it was found to be inconsistent with the working mechanism[7–9]. UCPL was also reported in novel materials including carbon nanotube[10], $CsPbBr_3$ perovskite[11], diamond[12], and monolayer $WS_2$[13], but their UCPL efficiencies are insufficient for practical applications. Overcoming most of the hurdles for organic dyes and rare-earth-doped materials, bulk semiconductors[14–17] are technically prohibitive for the crystals to reach an extremely low concentration of non-radiative recombination centres to achieve efficient UCPL under common excitation conditions[17]. Besides, bulk semiconductors with high refractive indexes will trap the emission light and cause strong reabsorption[14,15]. Colloidal quantum dots (QDs)—semiconductor nanocrystals with their sizes in the quantum confinement regime—have also been explored for UCPL[18–22]. The generally accepted mechanism of QD UCPL is based on thermal-activation of defect states within the bandgap[18,19,21], which implies low UCPL efficiency under typical excitation conditions. A few publications reported semiconductor nanomaterials with high power-conversion efficiency under sub-bandgap excitation[22–24], which have been under debate in the recent literatures[17,25,26].

## Results and discussion

**Up-conversion photoluminescence properties of ensemble QDs.** Monodisperse CdSe/CdS core/shell QDs with various core sizes and shell thicknesses are synthesized according to the literature[27] (see Supplementary Methods), yielding QDs with nearly ideal down-conversion photoluminescence (DCPL) properties (Supplementary Fig. 1). Figure 1a shows that the UCPL and DCPL spectra of a typical sample closely resemble each other (Supplementary Fig. 2). Slight red-shifting and narrowing of UCPL spectra are attributed to minor size inhomogeneity (Supplementary Fig. 3 and Supplementary Discussion). The UCPL intensity linearly depends on the excitation power density (Fig. 1a, inset), indicating an one-photon up-conversion process with the energy discrepancy compensated by phonons[21]. Both UCPL spectrum (Supplementary Fig. 4) and photoluminescence excitation spectrum (Fig. 1b) show no defect-related signals in the high wavelength part[18,28]. Photoluminescence excitation spectrum is found to overlap with the entire absorption spectrum nicely (Fig. 1b), implying equal photoluminescence quantum yield for UCPL and DCPL[29]. The photoluminescence excitation spectrum is monitored at the photoluminescence peak position to exclude the influence of size distribution. The absolute

photoluminescence quantum yield excited at 450 nm is measured as $0.99 \pm 0.01$ by an integrating sphere system (Supplementary Figs. 5–7). The UCPL quantum yield excited at 638 nm is determined to be unity taking the quantum yield excited at 450 nm of the same sample as reference (Supplementary Fig. 8), which indicates that the power-conversion efficiency of UCPL is above-unity for the QDs. The detailed methods of quantum yield measurements are shown in Supplementary Methods.

For both UCPL and DCPL, relaxation processes of the photon-generated carriers to form band-edge excitons are comparatively studied by femtosecond-resolved photoluminescence spectroscopy (Fig. 1c). Evidently, for DCPL (excitation wavelength shorter than 600 nm in the plot), the energy of the excitation photons determines the number of phonons coupled during the down-conversion relaxation, thus presenting a positive correlation with the relaxation time[30]. All carriers generated in the first-excitonic absorption band (600–650 nm), either 'hot' ones generated by down-conversion excitation at 600 nm or 'cold' ones generated by up-conversion excitation at 650 nm in Fig. 1c, relax with time constants comparable with the instrument response function (200 fs). Being several orders of magnitude faster than the typical rates of the thermal-activation from defect states (or intermediate states) within the bandgap[19,28], up-conversion occurs in the high-quality QDs is unlikely to share the same mechanism commonly proposed for QD UCPL, namely, thermal-activation of defect electronic states within the bandgap[18,19,21]. These results also exclude photo-induced intra-band transitions during the UCPL measurements, which would substantially increase the time constant[31].

Femtosecond-resolved absorption and resonant Raman spectra are measured to identify features of the phonon coupling in UCPL. When the sample was excited at 645 nm (Fig. 1d, red), recovery signals of the bleached first-excitonic state at 620 nm oscillate strongly, implying strong coherent phonon coupling[32] accompanying up-conversion excitation. Conversely, by excitation into higher excitonic states, oscillation is very weak and irregular (Fig. 1d, black). The frequency of the coherent vibration matches that of the longitudinal optical phonon of CdSe QDs measured by Raman spectroscopy (Fig. 1d, inset), indicating that the phonon-assisted photon-absorption[16,33,34] occurs within the CdSe core and away from the CdS shells. Furthermore, strong and instantaneous electron–phonon coupling upon up-conversion excitation in Fig. 1d suggests that the broad first-excitonic absorption band is associated with electron–phonon coupled states.

To further confirm the nature of the phonon-assisted absorption, temperature-dependent absorption of QDs is studied. Figure 2a (inset) shows that, as the temperature increases from 300 to 360 K, the first-excitonic absorption band of the QDs shifts towards low energy and the peak absorbance decreases, which should be a mixed result of bandgap narrowing (obeying Varshni's equation[35]), decrease of electron–hole envelope-wavefunction overlap, and increase of phonon population[12,13,28]. To isolate the last effect, Fig. 2a presents the first-excitonic-absorption-peak overlapped absorption spectra at various temperatures. The gradual and significant broadening at the low-energy side of the absorption band with increasing temperature is a strong indication of phonon-assisted absorption[33,34]. Quantitatively, Fig. 2b gives the temperature-dependent absorbance at the excitation energy of twice $E_{LO}$ of CdSe lower than the absorption peak ($E_{1S}$), where $E_{LO}$ is the longitudinal optical phonon energy. Temperature-dependent results in Fig. 2b can be well fitted with a Boltzmann distribution function $A e^{-\Delta E/k_B T}$, where $A$ is a factor related to transition probability, $k_B$ is the Boltzmann constant and $T$ is temperature. $\Delta E$ denotes energy difference between two states and

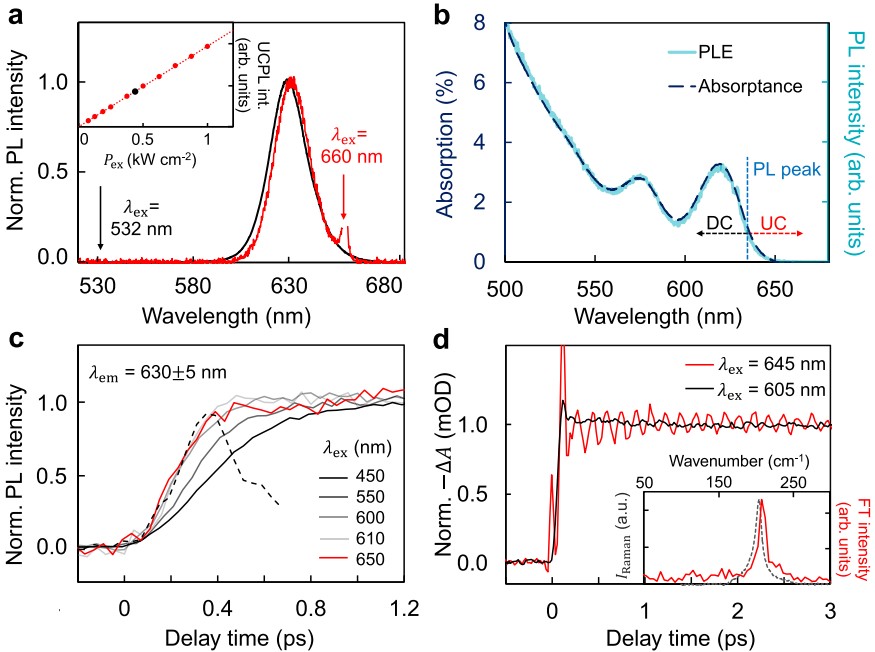

**Fig. 1 Spectral properties of ensemble QDs. a** Normalized UCPL (red) and DCPL (black) spectra excited at 660 and 532 nm respectively. $\lambda_{ex}$ denotes excitation wavelength. Inset: excitation power density ($P_{ex}$) dependent UCPL intensity. The red dashed line represents a linear fitting. The power density used in UCPL spectrum measurements is marked with a black dot. **b** Percentage absorption spectrum (dashed deep blue line) and photoluminescence excitation (PLE) spectrum (solid cyan line) monitored at the photoluminescence peak position (PL peak, 630 nm). 'DC' and 'UC' denote down-conversion and up-conversion excitation wavelength ranges respectively. **c** Normalized femtosecond-resolved photoluminescence spectra monitored at 630 nm under different excitation wavelengths. The black dashed line is the instrument response function. **d** Normalized femtosecond-resolved absorption spectra monitored at the first absorption peak (620 nm) under up-conversion (red) and down-conversion (black) excitations. Inset: the Fourier transform of the red curve in the main figure (red solid line) along with the resonant Raman spectrum of CdSe QDs (grey dashed line).

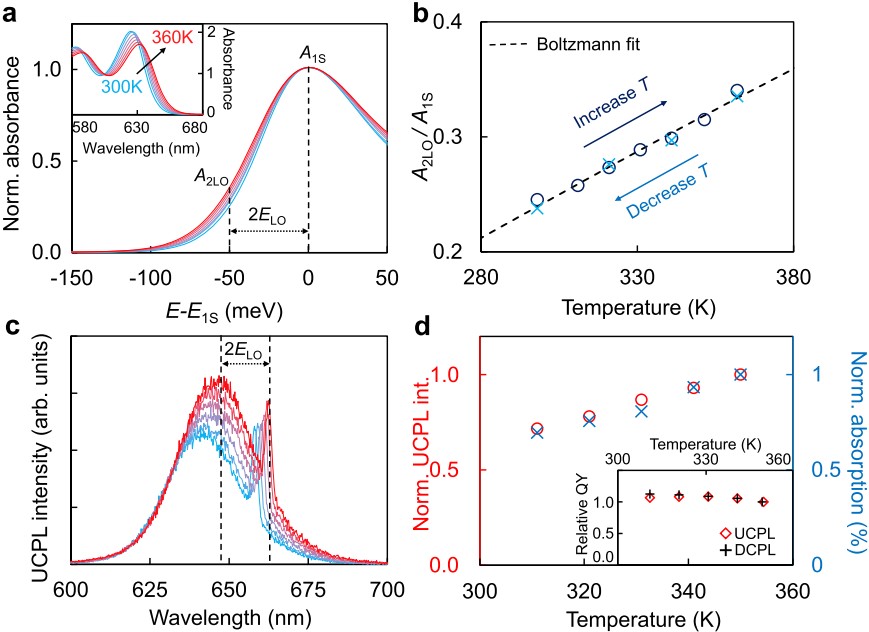

**Fig. 2 Temperature-dependent absorption and photoluminescence spectra of QDs. a** The first-excitonic absorption (1S) peak overlapped absorption spectra at different temperatures. The x-axis denotes the photon energy ($E$) relative to the energy of the 1S peak ($E_{1S}$) for each spectrum. $E_{LO}$ is the energy of the longitudinal optical phonon of CdSe. $A_{1S}$ and $A_{2LO}$ are the absorbances at $E_{1S}$ and $E_{1S}-2E_{LO}$ respectively. Inset: Original absorption spectra. **b** Temperature-dependent ratio of $A_{2LO}$ and $A_{1S}$ in both increasing (deep blue circle) and decreasing (light blue cross) temperature ($T$) processes. The black dashed line represents a fitting to the Boltzmann distribution function. **c** UCPL spectra at different temperatures with excitation photon energy being $2E_{LO}$ lower than the average photoluminescence photon energy. The corresponding normalized temperature-dependent UCPL intensity (red circle) and the percentage absorption at the excitation wavelength (blue cross) are plotted in (**d**). Inset: temperature dependence of relative quantum yields (QY) for UCPL and DCPL.

is determined to be 48 meV, in good agreement with twice $E_{LO}$ of CdSe (50 meV according to Fig. 1d). These results confirm that the absorption at the first-excitonic absorption peak of CdSe/CdS core/shell QDs is an electron–phonon coupled transition, with the longitudinal optical phonons of CdSe as major contributors.

To further exclude thermal-activated UCPL based on defect states within the bandgap, temperature-dependent UCPL is also recorded (Fig. 2c) with the excitation energies keeping twice $E_{LO}$ of CdSe lower than the UCPL peaks to eliminate the influence of spectral shift[10,35]. The peak shifts of DCPL and UCPL show identical temperature-dependence (Supplementary Fig. 9). Though UCPL intensity increases significantly with temperature, which is reported in various literatures[10,18,19,21], the temperature-dependence is found to follow exactly the same trend with that of the absorptance at the excitation wavelengths (Fig. 2d), confirming the temperature-independence of the UCPL quantum yields (Fig. 2d, inset). This result and the increase of band-tail absorbance with temperature discussed above are conflict with up-conversion based on thermal-activation of sub-bandgap defect states[28]. The DCPL quantum yield is also confirmed to be temperature-independent (Fig. 2d, inset).

**Up-conversion photoluminescence properties of single QDs.** Single-dot spectroscopy can directly demonstrate that DCPL and UCPL originate from the same emissive states by ruling out sample inhomogeneity as well as energy transfer and reabsorption. Figure 3a shows that the photoluminescence intensity trajectories of a representative single QD can be both nearly free of blinking—randomly switching of emission intensity between different brightness levels under constant excitation[36]—under either up-conversion or down-conversion excitation. For a typical QD, DCPL and UCPL spectra overlap well with each other (Fig. 3b), further confirming that the slight red-shifting and narrowing of UCPL spectra of QDs in ensemble level (Fig. 1a) is

not an intrinsic phenomenon, but caused by minor size inhomogeneity. Decay dynamics of both DCPL and UCPL follow the same mono-exponential function (goodness-of-fit $\chi_R^2$ smaller than 1.30) with a fitted lifetime of 24 ns (Fig. 3c), consistent with that of the ensemble measurements (Supplementary Fig. 10). While the above results confirm that the single-exciton emission states for DCPL and UCPL are indeed identical, the second-order photon correlation measurements imply the same bi-exciton quantum yield and decay dynamics for DCPL and UCPL (Fig. 3d). Statistically, unified single-dot DCPL and UCPL properties are observed (Supplementary Figs. 11–16). All measurements are carried out in the linear power-dependence range (Supplementary Fig. 17).

**Mechanism and applications.** The outstanding UCPL quantum yield and excellent up-conversion energy gain of the QDs encourage us to measure the optical cooling effects[2,4,5,15,17,22,37]. A 'QD-thermometer' is fabricated by filling QD solution into the tip reservoir of a quartz capillary tube. The sealed tube is then placed into a vacuum chamber with optical windows (Fig. 4a and Supplementary Fig. 18). The temperature change of the QD solution under laser irradiation is monitored according to the volume change of the solution (see Supplementary Methods). As control experiments, volume change of the solvent without QDs under the same experimental conditions is recorded. The temperature of QD sample and control specimen is calibrated with a water bath (Supplementary Figs. 19, 20). After subtracting the background heating of the control specimen, the QDs with emission peak at 632 nm excited by 450 and 532 nm lasers show strong and excitation-wavelength dependent heating effects (Fig. 4b and Supplementary Fig. 21). In sharp contrast, the heating effect is substantially reduced by excitation at 660 nm. When the excitation wavelength is further shifted to the tail of the absorption spectrum (671 nm), corresponding to an average

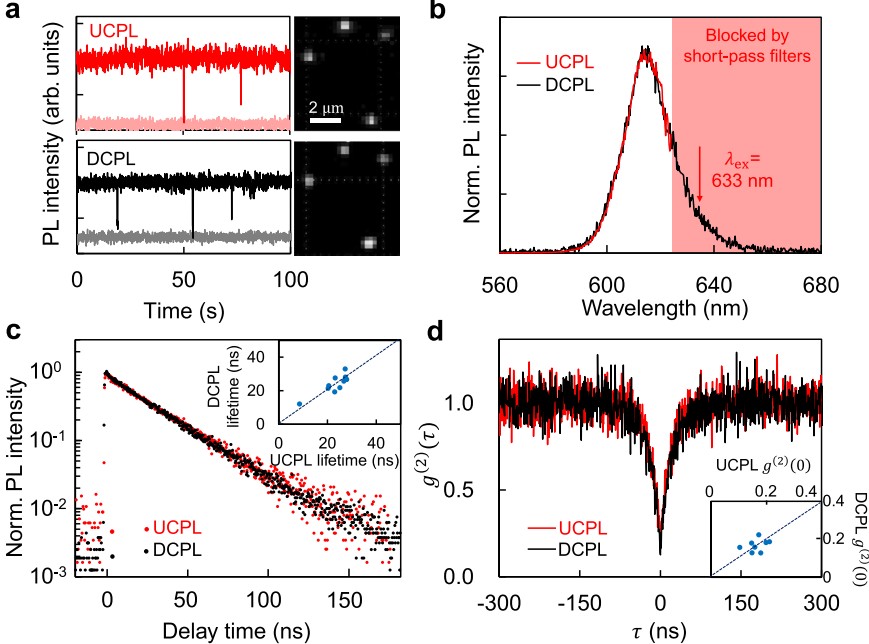

**Fig. 3 UCPL and DCPL properties of single QDs. a** UCPL and DCPL microscopic images of 4 single QDs (right) and representative intensity time trajectories of a single QD with backgrounds in light colours (left). **b** Normalized UCPL and DCPL spectra of a typical single QD. **c** Normalized decay dynamics for UCPL and DCPL of a typical single QD. Inset: correlation of UCPL and DCPL decay lifetimes of 11 single QDs. **d** Second-order correlation functions ($g^{(2)}(\tau)$) of a typical single QD under up-conversion and down-conversion excitations. Inset: correlation of UCPL and DCPL $g^{(2)}(0)$ of 11 single QDs. Dashed lines in the insets indicate equal values on $x$ and $y$ axes. The excitation wavelengths for UCPL and DCPL measurements are 633 and 405 nm, respectively.

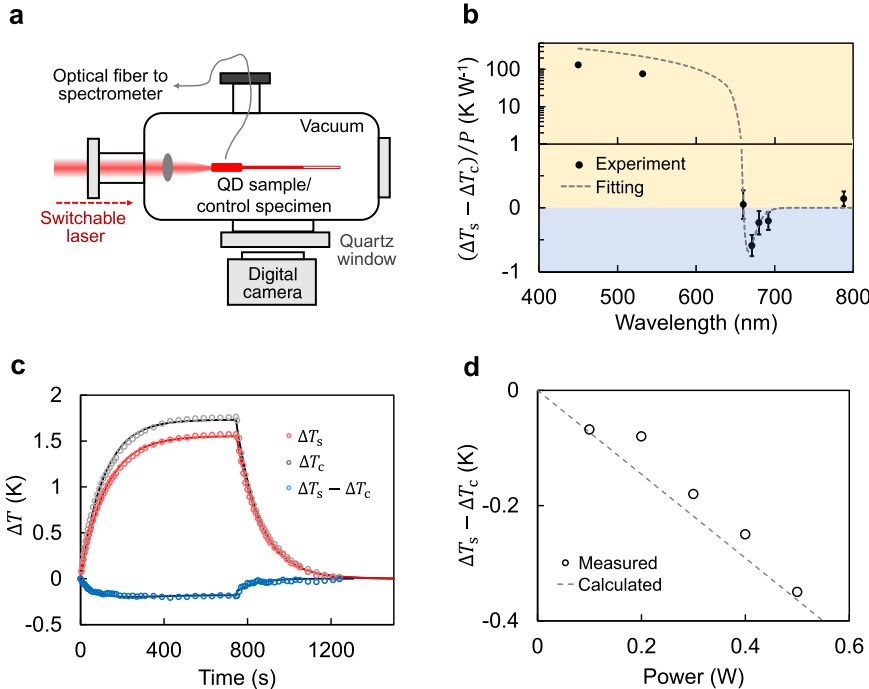

**Fig. 4 Optical cooling effects. a** Schematic diagram of the setup for optical cooling measurements. Temperature change of QD solution or control specimen in capillary tube under laser irradiation is determined by the volume change of the solution. **b** Relative temperature changes of the QD sample compared with the control specimen under different excitation wavelengths with unit excitation power (black dots). Error bars represent standard deviation based on three parallel tests. Dashed curve is a fitting according to our theoretical model (Eq. S31 in the Supplementary Information). The temperature changes are measured at equilibrium-states. **c** Temperature change kinetics of QD sample (red dots, $\Delta T_s$), control specimen (grey dots, $\Delta T_c$) and their difference (blue dots, $\Delta T_s - \Delta T_c$) measured at 671 nm. Solid lines are single exponential fittings (Eq. S22 and S23 in the Supplementary Information). **d** Excitation power dependent relative temperature change, measured at 671 nm and equilibrium-states. Calculated result (Eq. S31 in the Supplementary Information) is shown as dashed line.

up-conversion energy gain of 110 meV, a maximum temperature drop of 0.18 K under 300 mW irradiation relative to the control specimen is observed. Figure 4c shows the temporal evolution of temperature during the constant irradiation and after removal of the excitation (raw pictures are shown in Supplementary Fig. 22). When laser is turned on, both QD sample and control specimen are gradually heated but the temperature plateau of the QD sample is reproducibly lower than that of the control specimen (Supplementary Fig. 23), indicating a relative optical cooling for UCPL. The temperature change kinetics for both periods (constant irradiation and removal of excitation) are fitted with single exponential functions (solid lines in Fig. 4c), which gives a time constant of 121 (109) s for the QD sample (control specimen) at the 'laser on' stage and 109 (116) s for the QD sample (control specimen) at the 'laser off' stage. These values match reasonably well with the ones calculated based on a thermodynamic model (381s) with estimated thermal load and heat capacity (see Supplementary Methods). Furthermore, the equilibrium-state temperature increases near linearly with excitation power (Fig. 4d), being consistent with the model and the power-independent quantum yield of QDs (Fig. 1a, inset). Figure 4b shows that the equilibrium-state temperature changes under unit excitation power at different wavelengths including three up-conversion wavelengths (671, 680, and 692 nm) (Supplementary Fig. 21) can be well fitted with the theoretical model (see Supplementary Methods). When excited with photon energy far below the bandgap (788 nm), the temperature change of the QD sample is slightly larger than the control specimen, which can be attributed to the weak absorption of additional ligands and impurities in

the QD sample. Should the background absorption be reduced (by elaborately refine the media including ligands, solvents, and sample container), net cooling based on QDs should be achievable. Nevertheless, the relative cooling effects further confirm the near-unity quantum yield and phonon-coupling feature of UCPL of QDs.

The average up-conversion energy gain ($\Delta E_{\mathrm{avg}}$) of the QD-UCPL can be larger than 90 meV (Fig. 1a). The maximum energy gain ($\Delta E_{\mathrm{max}}$), which is defined as the energy difference between the highest detectable photoluminescence photon ($E_{\mathrm{PL}}$) and the excitation photon ($E_{\mathrm{ex}}$)[10,19,20,38], can be larger than 310 meV (Supplementary Fig. 24). These parameters and their relationship are illustrated along with the absorption and photoluminescence spectra in Fig. 5a. The energy gain values are much larger than either the energy difference between typical band-edge fine excitonic states (normally only several to tens of milli-electron volts)[39] or energy of a longitudinal optical phonon ($E_{\mathrm{LO}}$) of CdSe (25 meV) or CdS (38 meV)[40], indicating a multi-phonon-assisted UCPL process[13,28,34]. Though CdSe is a direct bandgap semiconductor[41], phonons can be involved in both absorption[16,33,34] and emission processes[42,43]. Figure 5a shows that the phonon-assisted absorption accounts for the energy gain of $E_{1S}-E_{\mathrm{ex}}$ (120 meV for the UCPL in Fig. 1a), which is slightly larger than $\Delta E_{\mathrm{avg}}$. For the electronic transition coupling simultaneously with $n$ phonons, the transition probability should be proportional to the $n$th power of the phonon population[28], which gives a calculated absorption spectrum shown in Fig. 5b (see Supplementary Methods for details), which semi-quantitatively matches the low-energy tail of the measured absorption spectrum. Figure 5b shows that, for the energy gain of $E_{1S}-E_{\mathrm{ex}}$ (120 meV, or

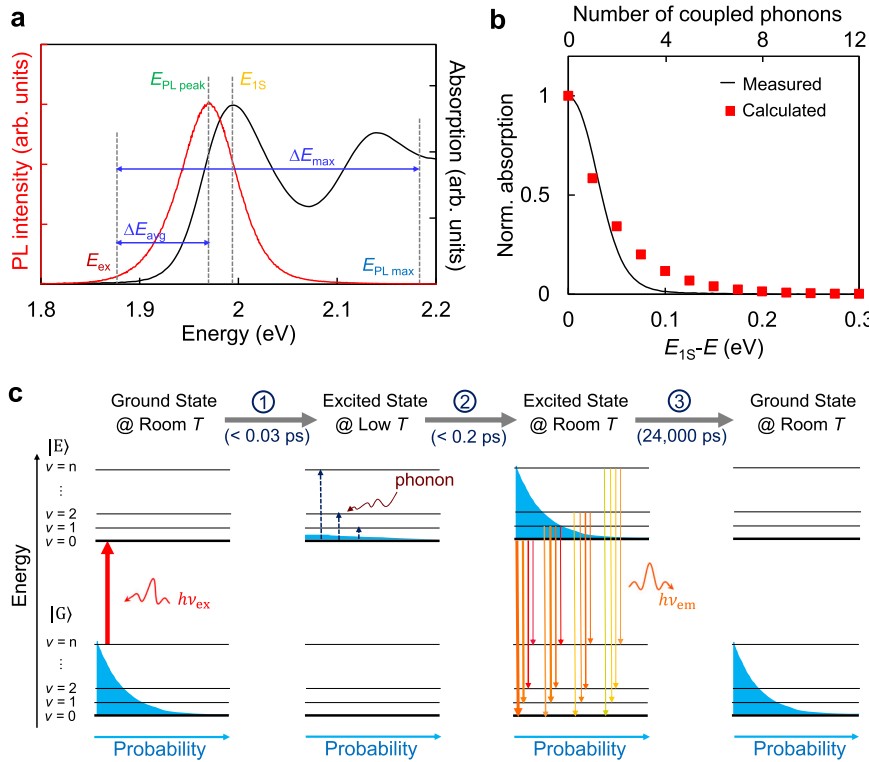

**Fig. 5 UCPL mechanism of high-quality QDs. a** Definitions of the average up-conversion energy gain ($\Delta E_{avg}$) and the maximum energy gain ($\Delta E_{max}$) for UCPL. $E_{ex}$, $E_{PL\ peak}$, $E_{1S}$, and $E_{PL\ max}$ are the energy of the excitation light, the photoluminescence peak, the 1st excitonic absorption peak and the maximum detected photoluminescence. The above energies are marked on the absorption (black) and photoluminescence (red) spectra. **b** Measured (black line) and calculated (red squares) absorption spectra normalized at the 1st excitonic absorption peak. The spectrum is calculated according to Eq. S33 in the Supplementary Information. $E_{1S}$–$E$ denotes the energy difference of the 1st excitonic absorption peak and the excitation photons. **c** Schematics of the UCPL mechanism. Shown is one set of electron–phonon coupled states with a specific phonon mode. $v$ is the quantum number of phonon. |G⟩ and |E⟩ are ground and excited states respectively. $hv_{ex}$ and $hv_{em}$ denote excitation and emission lights. Inter-band (intra-band) transitions are marked as solid (dashed) arrows.

coupled with five phonons), up-conversion excitation is experimentally and theoretically feasible.

Overall UCPL mechanism of the high-quality QDs is schematically summarized in Fig. 5c, which is thoroughly based on electron–phonon coupled intrinsic states[16,20,21,33]. At room temperature, electrons are thermally populated over the electron–phonon coupled states which are schematically drawn with one set of coupled phonon levels ($v$) for a specific mode (Fig. 5c, left). Within 0.03 ps (see Fig. 1d), low-energy photons excite the QDs at the room-temperature distribution of the electron–phonon coupled ground state (|G⟩) to the very bottom of its electron–phonon coupled excited state (|E⟩) (Fig. 5c, Process ①). This nearly instantaneous phonon-assisted absorption generates a transient state (Fig. 5c, the 2nd to the left), which represents a statistical thermodynamics 'low temperature' distribution on the phonon-coupled excited states. Consequently, the transient state relaxes to a 'room temperature' quasi-equilibrium distribution in a time constant smaller than 0.2 ps (Fig. 5c, Process ②). Finally, with barely any non-luminous defect for the band-edge exciton, the quasi-equilibrium distribution radiatively decays (time constant of nearly 24,000 ps) with unity quantum yield (Fig. 5c, Process ③), emitting a photon with energy higher than the excitation photon. As pointed out above, UCPL and DCPL share the same emission step (Fig. 5c, Process ③), namely, radiative decay from the room-temperature equilibrium distribution of the excited state to the room-temperature equilibrium distribution of the ground state. This means that, though the maximum energy gain ($\Delta E_{max}$) for UCPL—approximately the largest energy difference in the 3rd plot to the left in Fig. 5c—is partially realized in the emission step (Fig. 5a), this

part of phonon-coupled energy is shared for both UCPL and DCPL.

UCPL of QDs can be sufficiently bright for daily lighting under mild excitation. Excited by a 100-mW commercial AlGaInP light-emitting-diode (LED) peaked at 635 nm, monodisperse QDs filled in a quartz tube (Fig. 6a, inset) can produce bright orange UCPL, which expands the spectral width by nearly 100% to the short-wavelength side of electroluminescence (Fig. 6a). Owing to the readily-tunable UCPL, three pairs of LEDs and high-quality QDs (Fig. 6b) can offer nearly continuous white light with colour-rendering-index as high as 90.2 referring to blackbody-radiation at 3000 K accompanied by light-to-light power-conversion efficiency up to 104%.

Besides CdSe/CdS core/shell QDs demonstrated above, efficient UCPL is found to be universal among high-quality QDs with different sizes, crystal structures, morphologies, and compositions (see Supplementary Fig. 25, 26, and Supplementary Discussion). High-quality QDs have also been proven to be suited for realizing up-conversion electroluminescence with high power-conversion efficiency in QD-based LEDs fabricated with solution processes. It is anticipated that up-conversion electroluminescence coupled with up-conversion photoluminescence shall offer human being a new generation of light emission technology without waste heat yet in low cost.

## Methods

**Steady-state characterizations of ensemble QDs**. Transmission electron microscope characterizations were carried out using a Hitachi 7700 transmission electron microscope operating at 100 kV. Measurements of Raman spectra were carried out using a home-built inverted confocal microscope system. QDs were

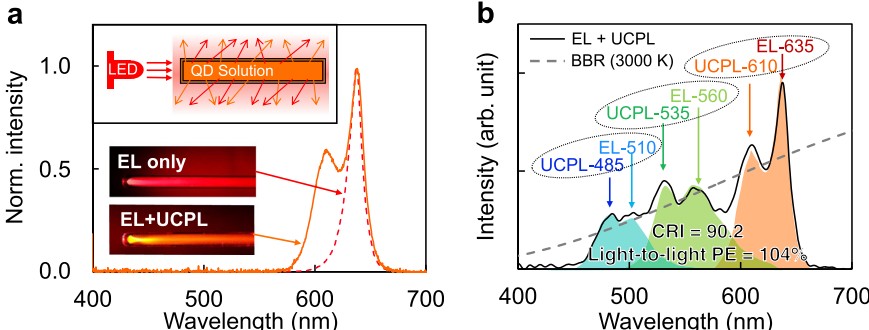

**Fig. 6 Application of QD-UCPL on lighting. a** Photographs and spectra of electroluminescence (EL) from a plain light-emitting diode (LED) (red dashed line) and EL + UCPL from a QD-based hybrid light source (orange solid line). Inset: schematic diagram for the hybrid light source. **b** Near-continuous spectrum of a hybrid white light source composed of three EL-UCPL units with a colour-rendering-index (CRI) of 90.2 referring to the blackbody-radiation (BBR) at 3000 K and a light-to-light power-conversion efficiency (PE) up to 104%. The peak wavelengths for each component are shown in the oval rings in nanometer.

dissolved in cycloheptane to exclude the Raman background of the solvent within the measurement wavelength range. The solution was then loaded in a quartz cuvette with 1 mm optical path. The cuvette was mounted on an Olympus IX83 inverted microscope with a ×20 objective (numerical aperture = 0.4). The excitation light source was a Cobolt 04-01-473 diode pumped solid-state laser with spectral peak position at 472.7 nm. The laser was focused onto the centre of the cuvette by the objective. The Raman scattering light was collected by the same objective and detected using a Princeton Instrument SP2750 monochromator equipped with a Pylon 400BRX liquid nitrogen cooled charge-coupled device (CCD) camera. A Semrock LP02-473RE 473 nm long-pass emission filter was used in front of the detector to block the strong Rayleigh scattering light. The system was calibrated using a clean silicon wafer. The solution was under continuous mechanical stirring during measurements. For all other optical characterizations, QDs were dissolved in toluene and measurements were performed using a quartz cuvette with 1-cm light path. The DCPL spectra presented in Fig. 5a, Supplementary Figs. 1, 3, 9, 25, 26 and the UCPL spectra presented in Fig. 2c, Supplementary Figs. 2, 3, 9, 25 were taken by the Edinburgh Instruments FLS920 spectrometer equipped with a 450 W Xe-lamp. For UCPL measurements, a 395 nm long-pass excitation filter was added in front of the sample to block the second-order diffraction light coexisting in the excitation light path. The DCPL and UCPL spectra presented in Fig. 1a and UCPL spectra in Supplementary Figs. 4, 24 were taken using a Princeton Instrument SP2750 monochromator equipped with a Princeton Instrument Pylon 400BRX liquid nitrogen cooled CCD. 532 and 660 nm narrow-band lasers were used as the excitation sources for DCPL and UCPL respectively. The excitation power was adjusted by a tunable neutral density filter. Absorption spectra of QDs were obtained using an Agilent Cary 4000 spectro-photometer. Photoluminescence excitation (PLE) spectra were taken by the Edinburgh Instruments FLS920 spectrometer operating on excitation scan mode. The detection wavelength was set as 630 nm (the emission peak position of the QDs). The photoluminescence quantum yield measurement methods are described in Supplementary Methods.

**Time-resolved characterizations of ensemble QDs**. For all time-resolved measurements, QD solutions were filled into quartz cuvettes with 1-mm light path and mounted on an Olympus IX83 inverted microscope equipped with a ×20 objective (numerical aperture = 0.4) and the sample solutions were under continuous mechanical stirring during measurements. For femtosecond-resolved photo-luminescence measurements, the output from a Light Conversion Pharos Yb:KGW laser (1030 nm, 200 fs, 1 MHz) was separated into two light beams. One was introduced to a Light Conversion Orpheus-HP optical parameter amplifier to generate a pump beam at a desired wavelength; the other passed through an OptoSigma OSMS26-300 optical delay line and used as a gating beam. During the measurements, the QD solution was irradiated by the pump beam. The photo-luminescence was collected with a concave mirror and adjusted to be collinear with the gating beam with a Thorlabs DMLP950 dichromatic mirror. The two beams were then focused onto a barium borate oxide (BBO) crystal to generate high-energy photons through a sum frequency generation (SFG) process. With a series of short-pass filters, the gating beam and the photoluminescence were blocked and the generated high energy photons were recorded using a Stanford Research System SR400 photon counter equipped with a Hamamatsu E717-500 photo-multiplier. The excitation wavelength for UCPL measurements of QDs was 650 nm and for DCPL measurements varied from 450 to 600 nm. The detection wavelength was set around 630 nm with a bandwidth of 10 nm for both DCPL and UCPL measurements. A Semrock BSP01-633R 633 nm short-pass filter was used right behind the sample to block the excitation light and the low-energy photo-luminescence photons in UCPL measurements. The time delay between the pump and the gating beams was adjusted by the optical delay line. The instrument

response function (IRF) of the whole system was measured by collecting the sum frequency signal generated by the scattered pump light and the gating light. The time constant of the rising edge of the IRF was determined to be 200 fs by fitting with a single exponential function. The full-width at half maximum (FWHM) of the IRF was 280 fs, giving an effective time resolution of 60 fs. For femtosecond-resolved absorption measurements, the fundamental output from another Light Conversion Pharos Yb:KGW laser (1030 nm, 30–40 fs, 100 kHz) was separated into two light beams. One was focused to an yttrium aluminium garnet (YAG) plate to produce a probe light with continuous spectra; the other was introduced to a Light Conversion ORPHEUS-N non-colinear optical parameter amplifier to generate a pump beam at a desired wavelength. The pump beam passed through a Newport M-ILS250HA optical delay line and a chopper working at 25 kHz driven by a Maxon EC motor, overlapping with the probe beam on the centre of the sample solution with a small angle. The transmitted probe light from the sample was then collected by a Zolix λ200i spectrometer. The pump beam was blocked using a polarization plate. The instrument response function of the system is about 30 fs by measuring the probe light intensity change of a blank sample.

**Measurements of temperature-dependent spectra**. Temperature control for the absorption spectra measurements of QD solution was carried out using an Agilent Cary SPV dual cell Peltier accessory. Temperature control for the photo-luminescence spectra measurements was carried out using a Julabo F12 GB heating-refrigeration water cycle bath equipped with a Julabo ED V.2 temperature controller. The accurate temperature of the QD solution was detected by a cali-brated LINI-T UT323 thermometer with its thermocouple immersed in the solu-tion. The spectra measurements at given temperatures were carried out in the same ways as described above.

**Photoluminescence characterizations of single QDs**. Photoluminescence char-acterizations of single QDs were carried out with a home-built inverted epi-fluorescence microscope system. QDs dissolved in toluene with 3% (weight/weight) of polymethyl methacrylate (PMMA) were spin-coated on a clean glass coverslip.

The sample was then mounted on a Princeton Instrument P-611.2 S piezo-stage which was further mounted on the sample stage of an Olympus IX83 inverted epi-fluorescence microscope. For DCPL measurements, the sample was excited with a PicoQuant PDL 800-D 405 nm laser through the objective in either continuous wave (CW) or pulsed mode for steady-state or transient photoluminescence measurements. In order to obtain a sufficient signal-to-noise ratio for single-dot UCPL measurements, the power of the excitation laser should be sufficiently high and the line width should be narrow. A HNJD-III 632.8 nm He-Ne laser was used as the excitation source for steady-state measurements and an Advanced Laser Diode Systems PiL063X 633 nm picosecond pulsed laser was used for transient measurements. In UCPL measurements, a Semrock LL01-633 band-pass excitation filter centred at 633 nm with 2 nm bandwidth was used. The excitation lights were reflected either by a Semrock FF614-SDi01 or a Semrock FF506-Di03 dichromatic mirror for UCPL and DCPL measurements respectively and excited the sample through an Olympus 60X oil immersion objective (numerical aperture = 1.49). Photons emitted by single QDs were collected with the same objective and directed to different exit ports of the microscope to acquire photoluminescence intensity trajectories, steady-state photoluminescence spectra, photoluminescence decay dynamics and second-order photon correlations respectively. For all UCPL measurements, a Semrock BSP01-633R 633 nm short-pass emission filter was used to block the strong excitation light. For effective excitation by the laser and avoiding blockage of the long-wavelength photoluminescence by the filters, CdSe/CdS core/shell QDs with photoluminescence peak position at 610 nm are

applied for single-QD UCPL measurements. The core size is 3.5 nm and the shell thickness is 3–4 monolayer. All measurements were performed at room temperature and ambient atmosphere. For DCPL measurements, the excitation power density on the sample was set to be 2 W/cm$^2$, which guaranteed the $<N>$ value, the average number of photons absorbed per QD per lifetime cycle, to be much smaller than 0.1, excluding generation of multi-excitons in QDs. For UCPL measurements, the excitation power density was set to be 50 W/cm$^2$ (Supplementary Fig. 17), yielding similar $<N>$ value for DCPL measurements. The photoluminescence signals were imaged and recorded by a Photometrics Evolve Delta electron multiplying charge-coupled device (EMCCD) camera with an integration time of 50 ms. The determination of 'on' and 'off' states is shown in Supplementary Discussion. The steady-state photoluminescence spectra of single QDs were recorded using an Andor 193i monochromator equipped with an Andor iXon DU-897U EMCCD. The power density for UCPL and DCPL measurements were 1 and 40 W/cm$^2$ respectively (Supplementary Fig. 17). In UCPL measurements, a series of short-pass emission filters were used to block the strong excitation light. The photoluminescence spectra were calibrated using an Alpha1501 halogen lamp. The photoluminescence decay dynamics was measured by a time-correlated single-photon counting (TCSPC) technique. The excitation peak power densities for DCPL and UCPL measurements were 4 and 60 kW/cm$^2$ respectively (Supplementary Fig. 17). The detector was a PicoQuant τ-SPAD APD. The time correlation was performed using a Becker & Hickl DPC-230 photo correlator. Second-order photon correlation measurements were performed using the same setup except the fluorescence was split with a 50/50 splitter and detected by two fibre-based APDs. The second-order photon correlation measurements reported in Fig. 3d were carried out with continuous-wave excitations. The excitation power densities used in all measurements were in the linear power-dependence region.

**Optical cooling experiment**. Optical cooling experiment was carried out using a home-built vacuum chamber system. For control specimen, toluene with 2% (m/m) oleylamine was injected into a thermometer-like quartz capillary attached to a bulb reservoir (Supplementary Fig. 18a). For QD sample, purified CdSe/CdS QDs were dispersed in toluene. 2% (m/m) of oleylamine was added to avoid ligand desorption. The concentrated solution (optical density being 1.2 at the first-excitonic absorption peak with 1 mm optical path) was filled into the same capillary tube used in control measurement (after cleaning). The control specimen was measured before the QD sample to avoid QDs remaining in the capillary tube. The inner diameter of the quartz capillary was only 0.16 mm while that of the reservoir was much larger (2 mm). The lengths of the capillary tube and the reservoir were 150 and 10 mm, respectively. The end of the capillary tube was sealed after the tube was partially filled with QD solution or solvent. The gas that remained in the tube is a mixture of air and saturated steam of toluene. The 'QD-thermometer' was then mounted on an adiabatic holder made of polylactic acid (Supplementary Fig. 18d). The holder was fixed on a three-dimensional manual stage which was placed in a vacuum chamber (Supplementary Fig. 18e). The air pressure in the chamber was kept lower than $4 \times 10^{-2}$ mbar under the working condition to diminish thermal convection with the atmosphere. The excitation light sources were narrow-band lasers with different peak wavelengths ranging from 450 to 788 nm. The spatial profiles of the excitation lights from different lasers were shaped to be identical with each other using an optical fibre and an iris diaphragm. The laser was directed into the chamber through a quartz window and irradiated the sample on the reservoir bulb (Fig. 4a). The excitation powers were adjusted with a neutral density filter and measured with a Thorlabs S121C power meter. The relative temperature changes were acquired by deducting the background heating of the control specimen. The temperatures of the QD sample and the control specimen were determined according to the volume change of the liquids. The details of temperature calibration and determination are described in Supplementary Methods. A set of typical raw data is shown in Supplementary Fig. 22.

**Fabrication of white light source**. For a specific EL-UCPL unit (Fig. 6a), the QD solution was filled into a quartz tube with an inner diameter of 2 mm. The tube was then mounted on a manual stage. The light source was a 100-mW commercial AlGaInP light-emitting-diodes (LEDs) with peak wavelengths being 635 nm and FWHMs being 17 nm respectively. The light emitted by the LED was focused, collimated, and irradiated the QD solution from the head of the quartz tube. UCPL of the QDs accompanied with the long-wavelength excitation light was detected by an Ocean Optics QE65000 Pro spectrometer through an optical fibre fixed at the side of the tube. In order to obtain enough excitation light escaping from the side of the tube, the incident light beam was deviated from the axis of the tube at a small angle. The intensity ratio of the UCPL and the unabsorbed excitation light can be adjusted by tuning the angle between the excitation light and the axis of the tube. A quartz diffuser was placed between the tube and the optical fibre to mix and uniformly distribute the UCPL and the excitation light. The continuous white light spectrum shown in Fig. 6b was obtained by combining the spectra of three EL-UCPL units with LED EL peaks at 510, 560, and 635 nm. The light-to-light power-conversion efficiency was calculated assuming that the photoluminescence quantum yield of three QD solutions were unity. The colour-rendering-index (CRI) was calculated according to the tristimulus values and spectra of 8 standard samples provided by International Commission on Illumination (CIE).

## Data availability
The data that support the finding of this study are available from the corresponding authors upon reasonable request.

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

## Acknowledgements

The authors thank Ms. Lili Huang (Zhejiang University, China) for the assistance in Raman spectroscopy measurements and Dr. Haiming Zhu (Zhejiang University, China) for providing the equipment for the femtosecond-resolved spectroscopy measurements. This work was supported by the National Key Research and Development Program of China (No. 2016YFB0401600) and the National Natural Science Foundation of China (No. 91833303).

## Author contributions

X.P. and H.Q. conceived the idea. Z.Y., X.L., and N.W. completed optical analysis, and J. Z., M.Z., and Z.Y. synthesized the QDs. X.P., H.Q., Z.Y., and X.L. wrote and revised the manuscript. All authors discussed the results and commented on the manuscript.

## Competing interests

The authors declare no competing interests.
