## [Peer Review File · Nature Communications]

REVIEWER COMMENTS

Reviewer #1 (Remarks to the Author):

In "Phonon-assisted up-conversion photoluminescence of quantum dots", Ye et al. report unity/near unity emission quantum yields (QYs) for CdSe/CdS nanocrystals (NCs) under both below and above band gap excitation. They suggest that up-conversion photoluminescence (UCPL) is mediated through electron-phonon coupled intrinsic states. The authors go on to try and demonstrate laser cooling with their high QY NCs. Overall, this study provides an interesting perspective into NC up-conversion and likely represents a good fit for the Journal. The reviewer, however, suggests major revisions to the manuscript. The reviewer's comments and suggested improvements are summarized below

Comments:

1. In the manuscript, the authors describe a three step model to explain the UCPL mechanism in CdSe/CdS NCs, in which electron-phonon coupled states assist sub-gap excitation (schematic diagram shown in Figure 4C). This model can explain UCPL with small up-conversion energy gains (i.e. the energy difference between the NC band edge and the sub gap excitation energy). Large up-conversion energy gain UCPL, however, cannot be explained by the model. Specifically, the authors claim that up-conversion energy gains as large as 310 meV are possible (Page 5, line 89). Given that LO phonon energies are of order 25 meV (Page 6, first line), electrons should interact with and gain energy from an excess of 8 phonons spontaneously to form such intrinsic states. It would seem that such higher order processes are extremely unlikely. To justify the appropriateness of the model proposed in the manuscript, the reviewer requests that the authors provide quantitative estimates of higher order phonon-mediated interband transition probabilities. These results should then be compared to obtained experimental results.

2. To estimate emission QYs, the authors measure absorption via sampling of excitation laser powers before and after samples with a power meter. Emission, by contrast, is measured using an integrating sphere. Because absorption and emission experiments are different, there will be measurement errors when data are combined to report an emission QY. This is described below.

a. In a standard QY measurement, involving an integrating sphere, excitation light is introduced into the sphere. This excitation light passes multiple times through the sample. Light is absorbed across these multiple passes. Emission then occurs and both residual excitation light and resulting emission light are measured on a spectrometer simultaneously through the sphere's output port. A differential spectrum that involves a solvent blank enables the ratio of integrated, emission-to-excitation areas to be linked to an absolute emission QY.

In the author's approach, absorption is measured on a separate instrument and considers only light that passes once through the sample. When this data is used together with the integrating sphere emission data, there will be a QY overestimate. This is because the author's approach underestimates the amount of light absorbed by samples, given that only one excitation pass occurs through the sample. Mixing and matching absorption and emission experiments will therefore lead to errors.

b. Continuing on this thought, there is an additional source of error in the absorption measurement that stems from emission emanating from samples. Namely, because the employed photodiode is not frequency selective, it will not distinguish emission from excitation signals. Consequently, apparent transmitted excitation powers will be larger than what is actually true given an additional emission contribution. This leads to a second underestimation of sample absorption. What likely results is a QY overestimate.

c. The reviewer thus requests that the authors provide a systematic error analysis emerging from points a and b above and provide that to readers in the Supplementary Material.

d. Then, to further validate the employed QY measurement approach, the authors should provide experimental data for organic dyes having known emission quantum yields.

e. Finally, the authors should directly compare their emission quantum yields (obtained using the stated approach) to values obtained exclusively using an integrating sphere. Any discrepancies should be explained.

3. Next, one of the more intriguing results of the paper is evidence for a relative temperature decrease of a toluene solution of NCs via sample volumetric changes. Provided is highly processed

data that the reviewer believes arises from a comparison of volumetric changes in the NC solution relative to that of a control toluene specimen. The authors suggest that because a volume difference exists between the two specimens under irradiation that relative cooling is achieved when NCs are present.

The reviewer finds this result intriguing. The reviewer, however, requests that the authors provide the raw data for sample and control volumetric changes. Volume calibration data for the control specimen should also be provided. Details of how control specimen volumes were calibrated for temperature should be provided. For example, how were control specimens heated and cooled? Details of the calculations used to link spectroscopic changes to specimen emission energies (via Varshni) to overall sample volumetric changes to eventual cooling powers should likewise be provided. Details of the sample preparation should also be presented. For example, are the solvents degassed prior to insertion into capillary tubes?

4. Continuing on this thought, in Figure 4B, there is only one data point to suggest that cooling has been achieved. The reviewer requests more data be taken so that error bars can be placed on all points. The reviewer also suggests that power- and below gap wavelength-dependent measurements be done for all cooling points.

5. The reviewer requests that the authors also show data for cooling and heating timescales. In effect, show data for the kinetics of cooling. Then turn off the sub gap excitation laser and let the sample warm up. Measure the associated heating kinetics. Characteristic cooling and heating timescales should then be compared to estimated cooling/heating times, based on the system's heat capacity and heat transfer parameters. In effect, the cooling/heating timescales should make sense.

Other comments:

1. For Figure 1a, the authors should provide the high wavelength part of the spectrum to show that there are no defect-related (i.e. deep-trap like) emission contributions to the data.

2. Page 3, line 39: Reference 2 by Rumbles shows up-conversion in a dye solution. However, the reviewer is concerned that the citation is not discussed within the wider context of laser cooling. As such, the introduction is misleading. Reference 2's claim of laser cooling is now widely regarded as false because claims in this paper have not been reproduced by other groups, despite numerous attempts. The reviewer therefore suggests that the introduction to this manuscript provide readers a more nuanced view of the current state of condensed phase laser cooling.

3. Page 11, line 17: Likewise, Reference 17 by Xiong represents another controversial paper that contains concerning claims about time scales required for heat transfer at nanometer length scales. The reviewer points to a recent discussion in Nature that highlights concerns that exist over Xiong's claims. The reviewer thus again requests that the authors provide readers a more representative and nuanced view of the current state of condensed phase laser cooling. As written, the introduction is misleading and implies that laser cooling for semiconductors has been achieved when, in fact, this is still highly debated.

4. Page 9, line 168: The authors refer to References 3 and 4 for potential suggestions for how to achieve net cooling with NC samples. Suggested sample refinements include changing surface ligands and solvent. References 3 and 4 are rare earth doped glasses and crystals. There are no ligands and solvents involved. The reviewer therefore suggests that the authors modify their sentence.

Reviewer #2 (Remarks to the Author):

In this work, Ye et al make core shell CdSe/ CdS QDs and compare the photoluminescence when excited at two different wavelengths, both at the single particle and ensemble level. In particular, excitation is at wavelengths above and below the absorption maxima of the nanoparticles. While excitation at energies above this maxima results in routine Stokes-shifted PL, excitation at energies below this maxima is what the authors call 'upconversion'. They argue that this upconversion PL is single phonon assisted with various temperature dependent measurements and power dependence. They also create a sophisticated QD thermometer and show a lot of data, including photon correlations of single QDs measured with CW excitation.

Reviewer #3 (Remarks to the Author):

This is a very interesting, important, and well-written manuscript which presents very well-designed experiments and corresponding data to support its conclusions. It shows that CdSe/CdS core-shell QDs can exhibit very efficient up-conversion PL (UCPL) that does not involve photoexcitation of occupied defect states within the semiconductor "bandgap" followed by thermal-activation of the carrier-occupied defects to generate carriers at their parent band-edge, followed ultimately by radiative band-band recombination. Instead, the UCPL is shown to originate from a multi-phonon assisted absorption process across the band gap, followed by coherent carrier-phonon coupling (viz. in the conduction band) to establish an energetic carrier- multiphonon distribution in the excited band (viz. conduction band), which then leads to a more energetic excited state band-band radiative transition, producing UCPL. In essence the Ground State (valence band) carrier-phonon coupled distribution is transferred by bandgap photons to the Excited State carrier-phonon distributions, and relaxation of the excited state produces upconverted photons emitted via PL. But I have two comments/questions the authors should consider in a minor revision:

1. CdSe is in its bulk a direct semiconductor requiring no phonon involvement, so why is the photoexcitation in CdSe QDs an indirect transition characterized by phonon assistance? This should be explained. I suggest the authors look at the paper in *J. Electronic Materials*, 28, No5, 414-425 (1999) to find answers that may apply here.
2. The model for the UCPL described here is not clearly described in a clear diagram. Fig 4c is stated to do so, but it is confusing and not clear. It should be redrawn to show more intermediate steps.

Point-by-Point Response to the Reviewers

Reviewer #1

General Comments: In “Phonon-assisted up-conversion photoluminescence of quantum dots”, Ye et al. report unity/near unity emission quantum yields (QYs) for CdSe/CdS nanocrystals (NCs) under both below and above band gap excitation. They suggest that up-conversion photoluminescence (UCPL) is mediated through electron-phonon coupled intrinsic states. The authors go on to try and demonstrate laser cooling with their high QY NCs. Overall, this study provides an interesting perspective into NC up-conversion and likely represents a good fit for the Journal. The reviewer, however, suggests major revisions to the manuscript. The reviewer’s comments and suggested improvements are summarized below.

Our revision and responses: We thank the reviewer for his/her positive feedback on our work. Extensive revisions are made to address his/her concerns (see below).

Comment 1: In the manuscript, the authors describe a three step model to explain the UCPL mechanism in CdSe/CdS NCs, in which electron-phonon coupled states assist sub-gap excitation (schematic diagram shown in Figure 4C). This model can explain UCPL with small up-conversion energy gains (i.e. the energy difference between the NC band edge and the sub gap excitation energy). Large up-conversion energy gain UCPL, however, cannot be explain by the model. Specifically, the authors claim that up-conversion energy gains as large as 310 meV are possible (Page 5, line 89). Given that LO phonon energies

are of order 25 meV (Page 6, first line), electrons should interact with and gain energy from an excess of 8 phonons spontaneously to form such intrinsic states. It would seem that such higher order processes are extremely unlikely. To justify the appropriateness of the model proposed in the manuscript, the reviewer requests that the authors provide quantitative estimates of higher order phonon-mediated interband transition probabilities. These results should then be compared to obtained experimental results.

Our revision and responses: We thank the reviewer for raising this critical issue. We shall manifest below that such large up-conversion gain can be explained by the phonon-assisted UCPL model.

By the definitions in the literatures (Wang, X. Y. et al. *Phys. Rev. B* **68**, 125318 (2003); Fernee, M. J. et. al. *Appl. Phys. Lett.* **91**, 043112 (2007); Akizuki, N. et al. *Nat. Commun.* **6**, 8920 (2015)), the average up-convention energy gain (ΔE_{avg}) is the energy difference between the average PL (the PL peak for QDs) and the excitation photon ($E_{\text{PL peak}} - E_{\text{ex}}$). While the maximum up-conversion energy gain (ΔE_{max}) is defined as the energy difference between the highest detectable photoluminescence photon and the excitation photon ($E_{\text{PL max}} - E_{\text{ex}}$). Experimentally, we determine $E_{\text{PL max}}$ as the photon energy where the photoluminescence intensity is 3 times the standard deviation (std DEV) of the background noise above the average background signals at the high energy side of the PL spectrum. Apparently, ΔE_{max} is related to the sensitivity of the detector. The above parameters and their relationship are illustrated in Fig. R1 as well as Fig. 5a in the revised main text.

Figure R1. Definitions of ΔE_{avg} and ΔE_{max} for UCPL. E_{1S} is the energy of the 1st excitonic absorption peak. Parameters are illustrated along with a typical PL spectrum and mechanism schematics.

The unexpected maximum energy gain (ΔE_{max}) of 310 meV is achieved through two independent steps: phonon-assisted absorption (Khurgin, J. B. *Appl. Phys. Lett.* **104**, 22115 (2014); Sun, G. et al. *ACS Photonics* **2**, 628-632 (2015)) and phonon-assisted emission (Empedocles, S. A. et al. *Phys. Rev. Lett.* **77**, 3873-3876 (1996); Cui, J. et al. *Nano Lett.* **16**, 289-296 (2016)), which means that all phonons are not gained simultaneously. **Phonon-assisted absorption** mainly accounts for the energy gain of $E_{1S} - E_{\text{ex}}$ (~ 5 LO phonons of CdSe in Fig. 1a, slightly larger than ΔE_{avg}), where E_{1S} is the energy of the 1st excitonic absorption peak. **Phonon-assisted emission** accounts the rest energy of ΔE_{max} ($E_{\text{PL max}} - E_{1S}$, ~ 8 LO phonons as pointed out by the reviewer).

Specifically, in the first step (phonon-assisted absorption), a sub-bandgap photon excites a QD from electronic ground-state to the electronic excited-state with energy compensated by phonons (Fig. R1, red arrow). As the reviewer pointed out, the transition probability decreases as the number of phonons involved increases. The phonon population (N_p) of a specific mode obeys Bose-Einstein statistics, as

$$N_p = \frac{1}{e^{E_p/k_B T} - 1} \quad (\text{Eq. R1})$$

where E_p is the phonon energy (~ 25 meV for the LO phonon energy of CdSe), k_B is the Boltzmann constant and T is temperature. The probability of the electronic transition coupling simultaneously with n phonons should be in proportion to the n^{th} power of the phonon population (Morozov, Y. V. et al. *ACS Energy Lett.* **2**, 2514-2515 (2017)).

$$P \propto (N_p)^n = \left(\frac{1}{e^{E_p/k_B T} - 1} \right)^n \quad (\text{Eq. R2})$$

We estimate the absorption probability in Fig. R2, which qualitatively matches the low-energy tail of the absorption spectrum. In Fig. 1a in the main text, we used a 660 nm laser to excite QDs with their 1st exciton absorption peak at 620 nm. Therefore, phonon-assisted absorption account for the energy gain of ~ 120 meV contributed by ~ 5 LO phonons. Based on Eq. R2, the excitation transition probability at 660 nm is 6.8% of that at 620 nm. The experimental result is 0.9%. The discrepancy should be attributed to the Franck-Condon factor (the overlap integral of the vibrational wave functions of the initial and final states) which decreases with increasing of the coupled phonons for materials with small Huang-Rhys factors, such as quantum dots.

Figure R2 (Figure 5b in the main text). Estimation of the phonon-assisted absorption probability. The absorption tail (black line) and the calculated results based on Eq. R1 (red square) are qualitatively matched. The discrepancy can be attributed to the Franck-Condon factor. The x-axis represents the photon energy relative to the energy of the 1st excitonic peak (E_{1S}). The blue dash line represents the transition involving 5 phonons.

Next, the carriers form a quasi-equilibrium distribution on the electron-phonon coupled states by interacting with phonons with relatively low energies, such as acoustic phonons. The phonon-assisted process occurs much faster than the exciton radiative decay. Consequently, energy of the emitted photons shall follow equilibrium distribution, some of which can be much larger than that of the bandgap by absorbing LO phonons (Fig. R1, green and blue arrows). This process accounts for the up-conversion energy gain of $E_{PL\ max} - E_{1S}$ (~ 190 meV or 8 excess phonons).

It is worth noting that the energy gain in phonon-assisted emission occurs in both down-conversion and up-conversion photoluminescence with identical value. According to measurements here, time constants of the thermo-equilibrium processes of the excitons

formed by either type of excitation are sub-picoseconds (Fig. 1c). The radiative decay (lifetime in tens of nanoseconds, Supplementary Fig. 10) for either type of excitations occurs from the same thermo-equilibrium distribution of excitons. In this sense, the energy gain of phonon-assisted emission contributed to the maximum energy gain (ΔE_{max}) is not really an “up-conversion”. Conversely, the energy gain in the phonon-assisted absorption exclusively occurs in up-conversion photoluminescence.

Comment 2: *To estimate emission QYs, the authors measure absorption via sampling of excitation laser powers before and after samples with a power meter. Emission, by contrast, is measured using an integrating sphere. Because absorption and emission experiments are different, there will be measurement errors when data are combined to report an emission QY. This is described below.*

Our revision and responses: We apologize for the confusion. Actually, we applied two independent QY measurements, and we did not use a power meter for absorption and an integrating sphere for emission in one quantum yield measurement.

The absolute quantum yield of the quantum dots excited at 450 nm was determined using an integral sphere system (Fig. R3a). Both the absorption and emission were determined according to the irradiance of the light exiting from the integrating sphere. The system was calibrated with a standard light source. The accuracy and reproducibility of the system has been confirmed by measuring the quantum yields of standard dyes, e.g. Rhodamine-6G and Nile Red (see below).

The UCPL quantum yield of the quantum dots excited at 638 nm was determined using a relative method, in which absorbance was measured by a power meter and photoluminescence was measured by a spectrometer. The quantum yield of the QDs excited at 638 nm was determined by taking the quantum yield of the same sample excited at 450 nm as reference. The accuracy of this method was confirmed by measuring both the absolute and relative quantum yields (0.97 and 0.98 respectively, see Supplementary Fig. 7 & 8) of the sample excited at 532 nm.

We use a relative method to determine the UCPL quantum yield, instead of using an integrating sphere system for absolute measurements. The absorbance of the sample excited at sub-bandgap wavelength is so low that the residual excitation light is much stronger than the photoluminescence. Furthermore, in terms of spectrum recording, sub-bandgap excitation and emission overlap with each other heavily. These facts make it difficult to accurately determine the absolute UCPL quantum yield of the sample using an integrating sphere system. In the relative method, collecting emission signal perpendicular to the excitation light minimizes the interference of the residual excitation light.

The detailed description of the measurements is stated in the revised version of Supplementary Method 2.

Figure R3 (Supplementary figure 5). Schematic diagrams of the setups for photoluminescence quantum yield measurements. a, Schematic diagram of the setup for absolute quantum yield measurements. LEDs with center wavelength at 450 nm (or 532 nm) are directed to excite the sample in the integrating sphere. Photoluminescence and the residual excitation light exiting from the integrating sphere are collected by an optical fiber and recorded with a calibrated spectrometer. **b**, Schematic diagram of the relative quantum yield measurements. The excitation light is shaped with an optical fiber and an iris. The number of photons absorbed by the sample is determined with a power meter placed in front of and behind the sample. The distance between the sample and power meter is larger than 0.2 m when measuring the transmitted light power, in order to avoid the interference from emission light. The photoluminescence light is collected perpendicular to the excitation light and recorded with a spectrometer.

Comment 2a: *In a standard QY measurement, involving an integrating sphere, excitation light is introduced into the sphere. This excitation light passes multiple times through the sample. Light is absorbed across these multiple passes. Emission then occurs and both*

residual excitation light and resulting emission light are measured on a spectrometer simultaneously through the sphere's output port. A differential spectrum that involves a solvent blank enables the ratio of integrated, emission-to-excitation areas to be linked to an absolute emission QY.

In the author's approach, absorption is measured on a separate instrument and considers only light that passes once through the sample. When this data is used together with the integrating sphere emission data, there will be a QY overestimate. This is because the author's approach underestimates the amount of light absorbed by samples, given that only one excitation pass occurs through the sample. Mixing and matching absorption and emission experiments will therefore lead to errors.

Our revision and responses: The absolute photoluminescence QY excited at 450 nm is exactly measured with the method recommended by the reviewer (see above).

Specifically, QD samples with different concentrations (including a solvent blank) are excited with a 450 nm LED. The emission and residual excitation light output from the integrating sphere are simultaneously recorded using a spectrometer. With the spectrally converted photon number, a linear fit gives the quantum yield (Fig. R4 & R5). Compared with QY measurement where only two data points (one sample under test and one solvent blank) are involved, this modified measurement scheme would give a higher reliability (see Supplementary Method 2 for details).

While for the UCPL quantum yield measurements, a relative method is used where both the excitation light at 450 and 638 nm pass the sample only once (Fig. R3b). And the detected photoluminescence only produced when the excitation light passed the sample.

Therefore, there should not be an overestimation of quantum yield in our approach.

Comment 2b: *Continuing on this thought, there is an additional source of error in the absorption measurement that stems from emission emanating from samples. Namely, because the employed photodiode is not frequency selective, it will not distinguish emission from excitation signals. Consequently, apparent transmitted excitation powers will be larger than what is actually true given an additional emission contribution. This leads to a second underestimation of sample absorption. What likely results is a QY overestimate.*

Our revision and responses: As mentioned above, no power meter was involved for the absolute QY measurements. During the relative QY measurements, **the power meter was placed at the path of laser and ~0.2 m away from the cuvette** when measuring the laser power propagating through the sample (Fig. R3b). The active detector area has a diameter of 9.7 mm which covers a solid angle of 1.85 msr from the exciting spot. The QD solution under test has an isotropic emission, and there will be less than 0.015% of the emitted light reaching the active area of the power meter. **Therefore, the underestimation of sample absorption due to contribution from sample photoluminescence is as low as 0.015% for the relative QY measurements, which can be ignored.**

Comment 2c: *The reviewer thus requests that the authors provide a systematic error analysis emerging from points a and b above and provide that to readers in the Supplementary Material.*

Our revision and responses: By clearly describing the methods of quantum yield measurements, there should be no error emerging from point a. The error emerging from point b was less than 0.015%, which is negligible. The details are included in the Supplementary Information as suggested by the Reviewer. Thanks.

Comment 2d: *Then, to further validate the employed QY measurement approach, the authors should provide experimental data for organic dyes having known emission quantum yields.*

Our revision and responses: Yes, we did so periodically to check our systems. We measured the absolute photoluminescence quantum yields of Rhodamine-6G and Nile Red with the integrating sphere system. The reported photoluminescence quantum yields of Rhodamine-6G and Nile Red are 0.94 (Fischer, M. et al. *Chem. Phys. Lett.* **260**, 115-118 (1996)) and 0.78 (Sarkar, N. et al. *Langmuir* **10**, 326-329 (1994)). **The average value of photoluminescence quantum yields of Rhodamine-6G and Nile Red over three tests were 0.92 ± 0.01 and 0.79 ± 0.01 , which match well with the reported value (Fig. R4).**

The relatively lower value of the photoluminescence quantum yield of Rhodamine-6G can be attributed to sample degeneration or environmental problems. These results validate the accuracy of our absolute QY measurement setup.

Figure R4 (Supplementary Figure 6). Absolute quantum yield measurements for standard dyes. a, b and c are experimental results for the determination of quantum yield of Nile Red for three parallel tests. d, e and f are experimental results for the determination of quantum yield of Rhodamine-6G for three parallel tests. N_{PL} and N_t denote the photon numbers of the photoluminescence and residual excitation light. The grey dashed lines represent linear fits to the experimental data (Eq. S5). The absolute slopes of the fitting give the measured quantum yields.

Comment 2e: Finally, the authors should directly compare their emission quantum yields (obtained using the stated approach) to values obtained exclusively using an integrating sphere. Any discrepancies should be explained.

Our revision and responses: The photoluminescence quantum yield of quantum dots excited at 450 nm was obtained exclusively using an integrating sphere with uncertainty of ~ 0.01 . The photoluminescence quantum yield of quantum dots excited at 532 nm obtained with the stated relative method is 0.98 (**Supplementary Fig. 8**), which is close to the value obtained exclusively using an integrating sphere (0.97, Fig. R5).

Figure R5 (Supplementary Figure 7). Absolute quantum yield measurements for quantum dots excited at a, 450 nm and b, 532 nm. N_{PL} and N_t denote the photon numbers of the photoluminescence and residual excitation light. The grey dashed lines represent linear fits to the experimental data. The absolute slopes of the fitting give the measured quantum yields.

Comment 3: *Next, one of the more intriguing results of the paper is evidence for a relative temperature decrease of a toluene solution of NCs via sample volumetric changes. Provided is highly processed data that the reviewer believes arises from a comparison of volumetric changes in the NC solution relative to that of a control toluene specimen. The authors suggest that because a volume difference exists between the two specimens under irradiation that relative cooling is achieved when NCs are present.*

The reviewer finds this result intriguing. The reviewer, however, requests that the authors provide the raw data for sample and control volumetric changes. Volume calibration data for the control specimen should also be provided. Details of how control specimen volumes were calibrated for temperature should be provided. For example, how were control specimens heated and cooled? Details of the calculations used to link spectroscopic changes to specimen emission energies (via Varshni) to overall sample volumetric changes to eventual cooling powers should likewise be provided. Details of the sample preparation should also be presented. For example, are the solvents degassed prior to insertion into capillary tubes?

Our revision and responses: Thanks for the suggestions. We modified the main text and the Supplemental Information according to the reviewer's requests. Specifically, the raw data for sample and control volumetric changes have been added in Figure R6 (the revised Supplementary Fig. 22). Volume calibration data for the control specimen have been added in Figure R7 & R8 (the revised Supplementary Fig. 19 & 20). Details of how control specimen volumes were calibrated for temperature have been added in the revised Supplementary Method 3. More details of the sample preparation have been given in the revised Method section in the main text.

In the original manuscript, we used PL peak position of QDs to link temperature and volumetric change of the QD sample. The control specimen was not directly calibrated for temperature. However, we used the same quartz tube for both control and sample measurements and the thermal expansion properties of the control and the sample should

be almost identical. Hence, it is reasonable to assume the temperature-volume relationship is the same for the control and the sample.

In order to establish the temperature-volume relationship directly and more strictly in the revised manuscript, we used a temperature-controlled water-bath (Fig. R7) to calibrate the temperature-volume relation for the control specimen and QD sample, respectively. We attached a marker with sharp edges on the quartz tube to indicate the position change of the liquid level. We put control specimen or QD sample (in a capillary tube) into a water bath with designated temperatures. The temperature of water bath was monitored using a calibrated thermometer and the volume change was acquired by taking pictures using a digital camera. To minimize errors, we used the same capillary tube for both control and sample measurements. The temperature-volume calibrations for control specimen and QD sample were carried out independently. The calibration curves of control specimen and QD sample were nearly identical (Figure R8), which is consistent with our original assumption. The calibration method using water bath gives the similar calibration result with the calibration method using PL peak position (Figure R9). The relationship between temperature change and PL peak position (obeying Varshni's equation) is shown in Supplementary Fig. 9.

In the revised manuscript, the sections correlating the PL peak position changes with the volume changes (original Supplementary Method 3 and original Supplementary Fig. 16) are deleted. Sections on calibrating the temperature-volume relationship using the water-bath method have been added as the revised Supplementary Method 3 and the revised Supplementary Fig. 19 & 20. The sample photo was re-taken as Supplementary Fig. 18a. The main text has also been modified accordingly (Page 9, Line 166-168).

Figure R6 (Supplementary Figure 22). Photos of QD sample and control specimen under 671 nm laser irradiation at 300 mW. The red spots are liquid-air interfaces in the capillary tubes. $\Delta L_{s,max}$ and $\Delta L_{c,max}$ are the maximum position changes of liquid level in the capillary tubes of QD sample and control specimen respectively.

Figure R7 (Supplementary Figure 19). Temperature calibration with a water-bath. **a**, Schematic diagram of the calibration setup. The length of the marker and the liquid column above the maker are denoted as L_m and L respectively. **b**, Photo of temperature calibration for the control specimen. **c**, Photo of temperature calibration for the QD sample. The quartz tube in **b** and **c** is the same one. The quartz tube is sealed before the calibration.

Figure R8 (Supplementary Figure 20). Temperature calibration results for **a**, the control specimen and **b**, the QD sample. L_m and L are the length of the marker and the liquid column above the maker respectively.

Figure R9. Calibration result comparison of ‘water bath’ (direct) method (black circle) and ‘PL peak position’ method (red cross) for the QD sample. Dashed lines are linear fittings.

Comment 4: *Continuing on this thought, in Figure 4B, there is only one data point to suggest that cooling has been achieved. The reviewer requests more data be taken so that error bars can be placed on all points. The reviewer also suggests that power- and below gap wavelength-dependent measurements be done for all cooling points.*

Our revision and responses: We thank the reviewer for his/her critical suggestions. We carried out a large number of additional experiments of optical cooling for QDs according to the reviewer’s requests. Specifically, **three parallel experiments were carried out for both QD sample and control specimen (Fig. R10) to verify the repeatability** and to give the error bars marked in the revised Fig. 4b. The values of both positive and negative error bars are the sums of standard deviations of the temperature changes of the QD sample and the control specimen at each excitation condition. Besides the lasers used in the original

manuscript (450, 532, 660, 671, 788 nm), two additional lasers with sub-bandgap wavelengths (680 and 692 nm) were used for the wavelength dependent experiments. Significant cooling effect relative to the background was observed at excitation wavelength of 671 nm (Fig. R11), while faint cooling effects were observed at 680 and 690 nm (Fig. R12 & R13).

The wavelength dependent heating/cooling results shown in Fig. R13 are fitted well with our model (Eq. S31 in the Supplementary Information). Power dependent measurements were carried out under excitation wavelength of 671 nm (Fig. R14). **A nearly linear relationship between the relative temperature-change and excitation power was observed,** which is again consistent with our model (Eq. S31 in the Supplementary Information) and the power-independent quantum yield of QDs (Fig. 1a, inset).

Figure R11, R13 and R14 have been added in the revised Main text as Fig. 4c, 4b and 4d. Figure R10 and R12 have been added in the revised Supplementary Information as Supplementary Fig. 23 and 21. The texts in the main text (Page 9 & 10) and the Supplementary Method 3 have been modified accordingly.

Figure R10 (Supplementary Figure 23). Repeatability evaluation for the optical cooling experiments. Three parallel tests were carried out excited with 671 nm at 300 mW.

Laser was turned on at 0 s and turned off at 750 s.

Figure R11 (Figure 4c in the Main text). Temperature change kinetics of QD sample (red dots, ΔT_s), control specimen (grey dots, ΔT_c) and their difference (blue dots, $\Delta T_s - \Delta T_c$) measured at 671 nm. Solid lines are single exponential fittings (Eq. S22 & S23 in the Supplementary Information).

Figure R12 (Supplementary Figure 21). Temperature changes of the QD sample and the control specimen under laser irradiation at different wavelengths other than 671 nm. All results are the average values of three parallel tests. The temperature change at 671 nm is shown in figure 4c.

Figure R13 (Figure 4b in the main text). Relative temperature changes of the QD sample compared with the control specimen under different excitation wavelengths with unit excitation power. Error bars represent standard deviation based on three parallel tests.

Dashed curve is a fitting according to our theoretical model (Eq. S31 in the Supplementary Information). The temperature changes are measured at equilibrium-states.

Figure R14 (Figure 4d in the main text). Excitation power-dependent relative temperature change, measured at 671 nm and equilibrium-states. Calculated results (Eq. S31 in the Supplementary Information) are shown as dashed line.

Comment 5: *The reviewer requests that the authors also show data for cooling and heating timescales. In effect, show data for the kinetics of cooling. Then turn off the sub gap excitation laser and let the sample warm up. Measure the associated heating kinetics. Characteristic cooling and heating timescales should then be compared to estimated cooling/heating times, based on the system's heat capacity and heat transfer parameters. In effect, the cooling/heating timescales should make sense.*

Our revision and responses: We thank the reviewer for suggesting an insightful analysis to further strengthen scientific quality of the work. The temporal evolution of temperature during the laser was turned on/off is provided in Fig. R11(the revised Fig. 4c) and Fig. R12

(the revised Supplementary Fig. 21). The heating and cooling time constants of both QD sample and control specimen excited at 671 nm were determined to be ~ 110 s by single-exponential fits (Eq. S22 & S23 in the Supplementary Information). The cooling/heating constant was calculated as 381 s (Eq. S24 in the Supplementary Information) by estimating the heat capacity and heat load of the system. Experimental and theoretical results matched semi-quantitatively with each other. The difference should be caused by underestimation of the heat load and overestimation of the heat capacity. All the analysis above is provided as Supplementary Method 4 in the revised manuscript.

Comment 6.1 (other comments): *For Figure 1a, the authors should provide the high wavelength part of the spectrum to show that there are no defect-related (i.e. deep-trap like) emission contributions to the data.*

Our revision and responses: We have provided the high wavelength part of the up-conversion photoluminescence spectrum as Fig. R15 (revised Supplementary Fig. 4). It can be clearly seen that there is no defect-related emission in wavelengths up to 900 nm. The main text is modified accordingly (Page 5, Line 69-71).

Figure R15 (Supplementary Figure 4). Long wavelength part of the up-conversion photoluminescence spectrum. No defect-related emission is found in wavelength up to 900 nm.

Comment 6.2: *Page 3, line 39: Reference 2 by Rumbles shows up-conversion in a dye solution. However, the reviewer is concerned that the citation is not discussed within the wider context of laser cooling. As such, the introduction is misleading. Reference 2's claim of laser cooling is now widely regarded as false because claims in this paper have not been reproduced by other groups, despite numerous attempts. The reviewer therefore suggests that the introduction to this manuscript provide readers a more nuanced view of the current state of condensed phase laser cooling.*

Our revision and responses: We thank the reviewer for his/her kind comments. Accordingly, we have revised the introduction part of our manuscript (Page 3, Line 41 to 44).

Comment 6.3: *Page 11, line 17: Likewise, Reference 17 by Xiong represents another controversial paper that contains concerning claims about time scales required for heat transfer at nanometer length scales. The reviewer points to a recent discussion in Nature that highlights concerns that exist over Xiong's claims. The reviewer thus again requests that the authors provide readers a more representative and nuanced view of the current state of condensed phase laser cooling. As written, the introduction is misleading and implies that laser cooling for semiconductors has been achieved when, in fact, this is still highly debated.*

Our revision and responses: We apologize for the misleading and appreciate the reviewer's suggestion about the introduction of laser cooling in our manuscript. We agree that the works about laser cooling in semiconductors are highly debated (Morozov, Y. V. et al. *Nature* **570**, E60-E61 (2019); Zhang, S. B. et al. *NPG Asia Mater.* **11**, 54 (2019)) and have not been repeated by other groups yet. Particularly, the cooling time-scale reported in Xiong's work raised great concerns. On the other hand, laser cooling of rare-earth doped materials has been widely accepted and the cooling temperature was reported to reach sub-100 Kelvin (Melgaard, S. D. et al. *Sci. Rep.* **6**, 20380 (2016)). We have added a brief introduction about the current state of condensed phase laser cooling in the introduction section (Page 3 & 4).

We should emphasize that laser cooling is not the main concern of our work. We are aiming at applying up-conversion photoluminescence (as well as up-conversion electroluminescence presented in the Reviewer Only Material) in photoelectric devices, such as lighting, display, and solar cells to improve their power conversion efficiencies. The revised figure 6 show an example of up-conversion photoluminescence applied in

lighting.

Comment 6.4: *Page 9, line 168: The authors refer to References 3 and 4 for potential suggestions for how to achieve net cooling with NC samples. Suggested sample refinements include changing surface ligands and solvent. References 3 and 4 are rare earth doped glasses and crystals. There are no ligands and solvents involved. The reviewer therefore suggests that the authors modify their sentence.*

Our revision and responses: We appreciate the reviewer's suggestion about the sentence modification. We have removed the original references (#3 and #4) and the sentence now reads *“Should the background absorption be reduced (by elaborately refine the media including ligands, solvents, and sample container), net cooling based on QDs should be achievable.”*

Reviewer #2. *In this work, Ye et al make core shell CdSe/ CdS QDs and compare the photoluminescence when excited at two different wavelengths, both at the single particle and ensemble level. In particular, excitation is at wavelengths above and below the absorption maxima of the nanoparticles. While excitation at energies above this maxima results in routine Stokes-shifted PL, excitation at energies below this maxima is what the authors call ‘upconversion’ . They argue that this upconversion PL is single phonon assisted with various temperature dependent measurements and power dependence. They also create a sophisticated QD thermometer and show a lot of data, including photon*

correlations of single QDs measured with CW excitation.

Our revision and responses: We thank the reviewer for his/her positive feedback. No action is needed.

Reviewer #3

Main comment: *This is a very interesting, important, and well-written manuscript which presents very well-designed experiments and corresponding data to support its conclusions. It shows that CdSe/CdS core-shell QDs can exhibit very efficient up-conversion PL (UCPL) that does not involve photoexcitation of occupied defect states within the semiconductor “bandgap” followed by thermal-activation of the carrier-occupied defects to generate carriers at their parent band-edge, followed ultimately by radiative band-band recombination. Instead, the UCPL is shown to originate from a multi-phonon assisted absorption process across the band gap, followed by coherent carrier-phonon coupling (viz. in the conduction band) to establish an energetic carrier- multiphonon distribution in the excited band (viz. conduction band), which then leads to a more energetic excited state band-band radiative transition, producing UCPL. In essence the Ground State (valence band) carrier-phonon coupled distribution is transferred by bandgap photons to the Excited State carrier-phonon distributions, and relaxation of the excited state produces upconverted photons emitted via PL. But I have two comments/questions the authors should consider in a minor revision:*

Our revision and responses: We appreciate the positive evaluation of our work from the

reviewer. No action is needed here.

Comment 1: *CdSe is in its bulk a direct semiconductor requiring no phonon involvement, so why is the photoexcitation in CdSe QDs an indirect transition characterized by phonon assistance? This should be explained. I suggest the authors look at the paper in J. Electronic Materials, 28, No5, 414-425 (1999) to find answers that may apply here.*

Our revision and responses: Thanks for the suggestion, this fact is now noted in the revised manuscript with citation of the reference (Page 11, Line 209 & 210). To our understanding, the paper (Williamson, A. J. et al. *J. Electron. Mater.* **28**, 414-425 (1999)) stated that free standing CdSe quantum dots are direct bandgap materials. As pointed in the paper, the direct gap material can be transformed to indirect gap material only with (1) strong quantum confinement that shifts the Γ point over other local minimum points in the k -space; (2) large lattice mismatch. Compared with CdSe quantum dots, epitaxially growing CdS would make quantum confinement weaker. And lattice mismatch between CdSe and CdS is quite small (4%). **These two points indicate that CdSe/CdS QDs are direct-gap semiconductors as well.**

For direct gap semiconductors, indirect (phonon-assisted) transition can take place as long as energy and momentum conservation are satisfied (Khurgin, J. B. *Appl. Phys. Lett.* **104**, 221115 (2014); Cui, J. et al. *Nano Lett.* **16**, 289-296 (2016)). For above bandgap excitation, strong direct excitation dominates absorption process and overwhelms the phonon-assisted indirect processes. However, for sub-bandgap excitation, direct excitation is forbidden due to energy conservation restriction, which helps to uncover the relatively

weak indirect process.

Furthermore, for quantum dots, absorption and emission of photons are mediated by excitons rather than free carriers. The momentum of excitons is not necessarily to be zero, so that one or more phonons can be needed for the conservation of momentum. Moreover, even if the momentum of excitons in quantum dots is equal to zero, excitons can still interact with phonons with zero momentum (phonons at the center of the Brillouin zone).

Comment 2: *The model for the UCPL described here is not clearly described in a clear diagram. Fig 4c is stated to do so, but it is confusing and not clear. It should be redrawn to show more intermediate steps.*

Our revision and responses: We thank the reviewer for pointing out the unclear description and confusing figure related to the UCPL model. According to the reviewer's suggestion, we redrew the diagram (Fig. R16) and re-wrote the corresponding text in the manuscript (Page 12, Line 219 to 240). In addition to modification of the main scheme diagram (original Fig 4c, revised Fig. 5c), two more plots (revised Fig. 5a and 5b) are added in the revised Fig. 5 to further illustrate the mechanism.

Figure R16 (Figure 5c in the main text). Schematics of the UCPL mechanism. Shown is one set of electron-phonon coupled states with a specific phonon mode. $h\nu_{ex}$ and $h\nu_{em}$ denote excitation and emission lights. Inter-band (intra-band) transitions are marked as solid (dashed) arrows.

REVIEWER COMMENTS

Reviewer #1 (Remarks to the Author):

In the revised manuscript "Phonon-assisted up-conversion photoluminescence of quantum dots", the authors have addressed most of the reviewer's original comments. The up-conversion mechanism, QY measurements and laser cooling studies are now more clearly explained than in the originally submitted version. Overall, the reviewer is of the opinion that the study is a good fit for publication in Nature Communications. The reviewer does, however, suggest two additional items that could improve the manuscript prior to publication. Suggested improvements are summarized below.

1. In Figure 1a, a red shift of the UCPL, relative to the DCPL, is observed. The authors attribute this shift to the effects of residual ensemble size distributions. In Figure 1b, though, the PLE spectrum matches the observed absorbance spectrum without a noticeable redshift. It stands to reason that if size distributions do impact UPCL spectra, they should also emerge in acquired PLE spectra, especially given that they are generally acquired on the red edge of emission spectra. Please clarify and rationalize this point.

2. In Supplementary 12, twelve single quantum dot UCPL and DCPL spectra are shown. Peak emission positions are around 610 nm. This energy far from others seen throughout the manuscript (e.g. Figure 1, Figure 2, Figure 3 and Supplementary Figure 1). The authors should clarify this discrepancy.

Reviewer #3 (Remarks to the Author):

Response by authors to comments by this reviewer are accepted. Manuscript can be published as is.

Point-by-Point Response to the Reviewers (corresponding to the revised manuscript)

Reviewer #1

General Comments: *In the revised manuscript “Phonon-assisted up-conversion photoluminescence of quantum dots”, the authors have addressed most of the reviewer’s original comments. The up-conversion mechanism, QY measurements and laser cooling studies are now more clearly explained than in the originally submitted version. Overall, the reviewer is of the opinion that the study is a good fit for publication in Nature Communications. The reviewer does, however, suggest two additional items that could improve the manuscript prior to publication. Suggested improvements are summarized below.*

Our revision and responses: We thank the reviewer for his/her positive feedback on the first revision. Additional revisions are made to address his/her concerns (see below).

Comment 1: *In Figure 1a, a red shift of the UCPL, relative to the DCPL, is observed. The authors attribute this shift to the effects of residual ensemble size distributions. In Figure 1b, though, the PLE spectrum matches the observed absorptance spectrum without a noticeable redshift. It stands to reason that if size distributions do impact UPCL spectra, they should also emerge in acquired PLE spectra, especially given that they are generally acquired on the red edge of emission spectra. Please clarify and rationalize this point.*

Our revision and responses: We thank the reviewer for raising this issue. We agree with the reviewer that size-distribution can cause PLE spectrum shift and as we know, PLE

measurements are commonly used to evaluate size mono-dispersity of QDs. When monitored at the long-wavelength (short-wavelength) part of the PL spectrum, QDs with larger (smaller) sizes contribute more for the PLE spectrum, and thus noticeable redshift (blueshift) of the PLE spectrum compared with the absorption spectrum can be observed. **However, for PLE measurements in Fig. 1b, PL was monitored at the emission peak position (630 nm) with nearly no bias for QD size, given the very narrow size distribution of the sample. This makes the shift of PLE spectrum comparing to the corresponding absorption spectrum negligible.**

The interpretation of Fig. 1b in main text was modified accordingly, and it now reads “Photoluminescence excitation spectrum is found to overlap with the entire absorption spectrum nicely (Fig. 1b), implying equal photoluminescence quantum yield for UCPL and DCPL. **The photoluminescence excitation spectrum was monitored at the photoluminescence peak position to exclude the influence of size-distribution.**” (Page5, Line 73-75).

We have also described the PLE measurements in more detail in the Method section of the revised main text and it now reads “Photoluminescence excitation (PLE) spectra were taken by the Edinburgh Instruments FLS920 spectrometer operating on excitation scan mode. **The detection wavelength was set as 630 nm (the emission peak position of the QDs).**” (Page 15, Line 291-292).

Comment 2: *In Supplementary 12, twelve single quantum dot UCPL and DCPL spectra are shown. Peak emission positions are around 610 nm. This energy far from others seen throughout the manuscript (e.g. Figure 1, Figure 2, Figure 3 and Supplementary Figure 1). The authors should clarify this discrepancy.*

Our revision and responses: We thank the reviewer for raising this issue. The reason why we chose QDs with peak emission positions around 610 nm rather than 630 nm in single-QD measurements is to **match the wavelength of the excitation light source.** In single-QD measurements, the power of the excitation light source should be high enough to generate UCPL with sufficient signal-to-noise ratio. And the line width of the excitation light source should not be too large so that the excitation light can be fully blocked by filters. Therefore, we used a 633 nm He-Ne laser and a 633 nm pico-second pulsed laser as excitation light sources. A 633-nm short-pass emission filter was used to block the strong excitation light. According to the lasers used in the single-QD measurement, the peak emission position of QDs should be much shorter than 633 nm. Narrow-band pulsed laser with high output power and longer emission wavelength was hard to obtain then. Femto-second laser pulses generated from an optical parameter amplifier can reach wavelengths larger than 633 nm, but the band-width is too large (6-7 nm) so that the laser pulses can't be fully blocked by the short-pass emission filter.

The QDs used in single-QD measurements are also CdSe/CdS core shell QDs synthesized with the same method as the QDs used in ensemble measurements (see Supplementary

Method 1). The core size is the same (~3.5 nm) while the shell thickness is 3-4 monolayers instead of 7-8 monolayers. As pointed out in the last paragraph in the main text, the up-conversion photoluminescence is found to be universal for QDs with different sizes, compositions and crystal structures (see Supplementary Fig. S25). Besides, general optical properties including quantum yield, decay dynamics and stability are comparable for QDs with different shell thicknesses (Zhou, J. H. et al, *J. Am. Chem. Soc.* **139**, 16556-16567, (2017)). Therefore, the experimental results acquired from thin shell QDs should be universal.

The detail of the specific sample for single-dot UCPL is further given in Method section of the revised main text. The sentences now read “In order to obtain sufficient signal-to-noise ratio for single-dot UCPL measurements, the power of the excitation laser should be sufficiently high and the line width should be narrow.” (Page 18, Line 357-359) and “For effective excitation by the laser and avoiding blockage of the long-wavelength photoluminescence by the filters, CdSe/CdS core/shell QDs with photoluminescence peak position at 610 nm are applied for single-QD UCPL measurements. The core size is 3.5 nm and the shell thickness is 3-4 monolayer.” (Page 19, Line 372-376).

Accordingly, the sample structure information of QDs used in ensemble measurements has also been provided in the revised main text, and the sentence now reads “Monodisperse CdSe/CdS core/shell QDs with various core sizes and shell thicknesses are synthesized according to the literature” (Page 4, Line 61). And the UCPL and DCPL spectra in Figure

1a are specified as “the UCPL and DCPL spectra of a typical sample” (Page 4, Line 64-65).

Reviewer #3

General Comments: *Response by authors to comments by this reviewer are accepted.*

Manuscript can be published as is.

Our revision and responses: We thank the reviewer for his/her positive feedback. No action is needed.